# Reliability Scheduling for Robust Domain Adaptation under Label Noise

## Abstract

Unsupervised Domain Adaptation (UDA) becomes especially challenging when source labels are noisy, a situation common in real-world pipelines involving crowdsourcing or automated annotation. Label noise and domain shift jointly cause reliability issues: corrupted labels mislead supervision at the sample level, while ambiguous predictions hinder class-level alignment. Existing methods often address these issues in isolation with static heuristics, leading to fragile adaptation under severe noise. We introduce the **Reliability Scheduling Framework (RSF)**, which unifies noisy-label learning and domain adaptation through multi-scale reliability scheduling. At the sample level, **Confidence-Modulated Adaptive Learning (CMAL)** dynamically adjusts gradients using an entropy-guided exponent, suppressing noise memorization while retaining strong signals from reliable samples. At the class level, **Entropy-Guided Confusion Alignment (EGCA)** reweights alignment based on prediction entropy, reducing inter-class confusion and sharpening decision boundaries. Together, CMAL and EGCA coordinate how much to learn and what to align, yielding robust transfer even under heavy label corruption. Extensive experiments on Office-31, Office-Home, and VisDA demonstrate that RSF consistently outperforms prior state-of-the-art methods across symmetric and asymmetric noise settings. These results establish RSF as a principled and effective solution for robust UDA with noisy supervision.

## 1 Introduction

Deep neural networks have achieved remarkable progress in visual recognition, but their performance often collapses when training and deployment domains diverge. Unsupervised Domain Adaptation (UDA) aims to bridge this gap by transferring knowledge from labeled source data to unlabeled target data (Liu et al., 2022). While effective, most UDA approaches assume perfectly clean source labels. In practice, however, labels are frequently corrupted due to annotation errors, crowdsourcing, or automated pipelines, making this assumption unrealistic.

The presence of label noise creates a dual reliability challenge. On the one hand, corrupted supervision causes networks to memorize noisy labels (Li & Zhu, 2024), resulting in unreliable domain-invariant features. On the other hand, distribution shift exacerbates misalignment, as ambiguous target predictions near decision boundaries further propagate errors. Existing robustness-oriented approaches (Jin et al., 2024; Feng et al., 2023) rely on static heuristics such as fixed reweighting or uniform alignment. These designs overlook two critical aspects: (i) *sample-level reliability*, since not all source instances are equally trustworthy; and (ii) *class-level reliability*, since not all target predictions are equally informative. Without dynamically handling both dimensions, UDA remains vulnerable under severe noise.

We propose the **Reliability Scheduling Framework (RSF)**, which unifies noisy-label learning and domain adaptation into a curriculum-style paradigm of multi-scale reliability scheduling (Figure 1). At the *sample level*, RSF adjusts gradient strength according to prediction confidence, emphasizing reliable supervision and suppressing corrupted labels. At the *class level*, RSF regulates alignment based on predictive entropy, reducing inter-class confusion and sharpening decision boundaries. These two processes act complementarily: sample-level scheduling controls *how much to learn*, while class-level scheduling determines *what to align*. To instantiate RSF, we design two mechanisms. **Confidence-Modulated Adaptive Learning (CMAL)** introduces a reliability-aware

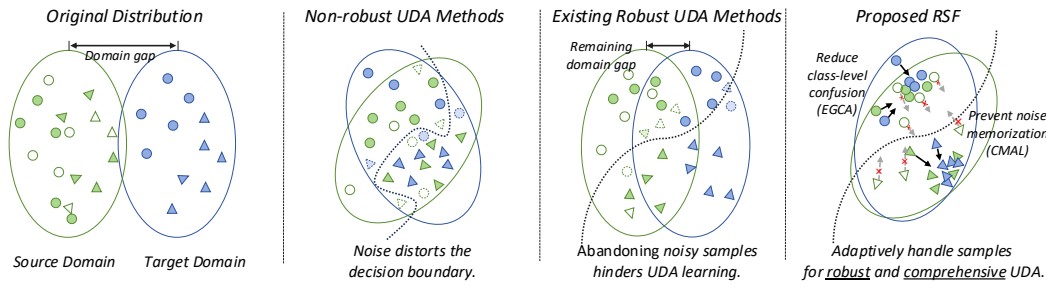

Figure 1: Comparison between prior UDA methods and our proposed framework. Unlike approaches that statically align domains regardless of label quality, our Reliability Scheduling Framework dynamically calibrates sample- and class-level reliability to enable robust adaptation under noisy labels.

loss with an entropy-guided exponent, preventing noise memorization while retaining strong signals from clean samples. **Entropy-Guided Confusion Alignment (EGCA)** defines an entropy-weighted alignment objective that prioritizes low-uncertainty target samples, ensuring reliable domain alignment. Together, CMAL and EGCA implement RSF as a principled and adaptive framework that achieves robust transfer under diverse noise conditions.

In summary, our contributions are threefold. (1) We introduce RSF as a new perspective on UDA with noisy labels, formulating it as a problem of multi-scale reliability scheduling. (2) We propose CMAL and EGCA as complementary mechanisms for sample- and class-level scheduling. (3) We conduct extensive experiments on Office-31, Office-Home, and VisDA, showing that RSF consistently outperforms state-of-the-art methods under varying noise levels, validating its robustness and generality.

Most existing UDA methods treat label noise and domain alignment as independent challenges, often using *static heuristics* such as fixed thresholds or uniform weighting. These strategies fail to adapt to the evolving reliability of samples and classes during training, leaving models prone to memorizing noisy labels and misaligning ambiguous target data.

We propose the **Reliability Scheduling Framework (RSF)**, which rethinks noisy UDA as a multi-scale scheduling problem. Instead of relying on handcrafted rules, RSF coordinates supervision and alignment through complementary mechanisms at two levels: *sample-level scheduling* and *class-level scheduling*. Concretely, RSF is instantiated with **Confidence-Modulated Adaptive Learning (CMAL)** for robust supervision and **Entropy-Guided Confusion Alignment (EGCA)** for uncertainty-aware alignment.

## 2 METHOD

### 2.1 PROBLEM DEFINITION

We study UDA under noisy source supervision. Let $\mathcal{D} = \{I_s, I_t\}$ denote a $C$-class source–target dataset, where $I_s = \{\mathcal{X}_s, \mathcal{Y}_s\} = \{(\boldsymbol{x}_i^s, y_i^s)\}_{i=1}^{N_s}$ is the source domain with corrupted labels, and $I_t = \{\mathcal{X}_t\} = \{\boldsymbol{x}_j^t\}_{j=1}^{N_t}$ is the unlabeled target domain. A domain-shared network $f = [f_1, \ldots, f_C]$ produces class probabilities with temperature $\tau$:

$$\hat{y}_{ij}^s = \frac{\exp(f_j(\boldsymbol{x}_i^s)/\tau)}{\sum_{k=1}^{C} \exp(f_k(\boldsymbol{x}_i^s)/\tau)}, \quad \hat{y}_{ij}^t = \frac{\exp(f_j(\boldsymbol{x}_i^t)/\tau)}{\sum_{k=1}^{C} \exp(f_k(\boldsymbol{x}_i^t)/\tau)}. \tag{1}$$

The goal is to mitigate *sample-level unreliability* from noisy labels and *class-level unreliability* from domain shift *jointly*.

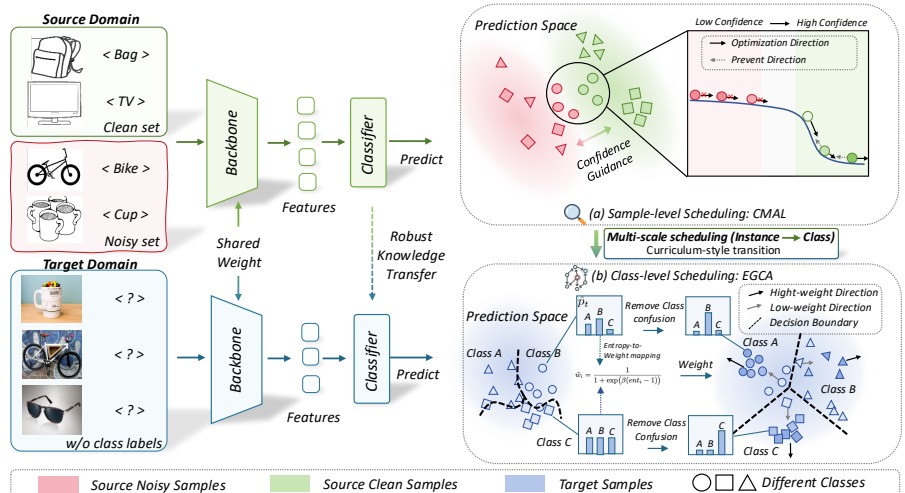

Figure 2: Overview of the proposed RSF framework. The model processes source domain data (with both clean and noisy labels) and target domain data (unlabeled) through a shared feature extractor. The framework incorporates two key mechanisms: (1) **CMAL** implements a dual-directional learning strategy. The optimization direction guides learning from all samples to prevent underfitting, while the prevention direction adaptively controls learning intensity. For high-confidence predictions, it preserves strong learning signals, while for low-confidence predictions (potential noisy labels), it reduces gradient magnitude to limit their negative impact. (2) **EGCA** utilizes entropy information to guide domain alignment, focusing on maintaining clear decision boundaries and reducing class-level confusion in the prediction space. These complementary mechanisms are optimized through $\mathcal{L}_{CMAL}$ and $\mathcal{L}_{EGCA}$ to achieve robust domain adaptation under noisy conditions.

## 2.2 Overview: Reliability Scheduling Framework

RSF performs **multi-scale reliability scheduling**: (1) *Sample-level scheduling* robustifies supervision by down-weighting dubious instances while preserving informative gradients from reliable ones. (2) *Class-level scheduling* regulates alignment strength based on predictive entropy, suppressing inter-class confusion and sharpening decision boundaries in the target prediction space.

This unified view explains why uniform reweighting or indiscriminate alignment often collapses under severe noise, and positions our two mechanisms as principled and complementary components.

## 2.3 Sample-level Scheduling: CMAL

We propose **Confidence-Modulated Adaptive Learning (CMAL)**, a reliability-aware loss that stabilizes training under label noise by modulating the contribution of each instance with a constant exponent $q \in (0, 1)$. For a source sample $(\boldsymbol{x}_i^s, y_i^s)$ with predicted probability $p_i \triangleq \hat{y}_{i,y_i^s}^s$, we define:

$$\mathcal{L}_{\text{CMAL}}(p_i) = \big(-\log p_i\big)^q \cdot (1 - p_i)^{1-q}, \qquad q \in (0, 1). \tag{2}$$

**Gradient analysis.** For $\mathcal{L}_{\text{CMAL}}(p) = (-\log p)^q (1-p)^{1-q}$ with $q \in (0, 1)$, the derivative w.r.t. $p$ is

$$\frac{\partial \mathcal{L}_{\text{CMAL}}}{\partial p} = \left[ -\frac{q(1-p)}{p} + (1-q)\log p \right] (-\log p)^{q-1} (1-p)^{-q}. \tag{3}$$

This gradient exhibits two characteristic regimes: (i) **Low confidence** ($p \to 0^+$). Since $p \log p \to 0$ and the $\frac{1}{p}$ term dominates,

$$\frac{\partial \mathcal{L}_{\text{CMAL}}}{\partial p} \sim -\frac{q}{p} (-\log p)^{q-1}.$$

Compared with the CE gradient $\partial(-\log p)/\partial p = -1/p$, CMAL introduces the multiplicative factor $(-\log p)^{q-1} \to 0$ (as $q - 1 < 0$). Thus the gradient still grows as $O(1/p)$, but with a significant

damping effect that moderates the explosion and prevents excessive updates on extremely noisy samples. (ii) **High confidence** ($p \to 1^-$). Using $\log p \simeq -(1-p)$, the bracket term in equation 3 satisfies

$$-\frac{q(1-p)}{p} + (1-q)\log p \;=\; -(1-p) + o(1),$$

while the prefactor behaves as $(-\log p)^{q-1}(1-p)^{-q} = (1-p)^{-1} + o((1-p)^{-1})$. Hence the overall gradient converges to $\frac{\partial \mathcal{L}}{\partial p} \to -1$, exactly matching CE at $p \to 1$. This ensures CMAL maintains a non-vanishing gradient signal to consolidate reliable supervision.

*Remark.* Strictly speaking, CMAL does not eliminate the gradient explosion in the limit $p \to 0$, but the $(-\log p)^{q-1}$ factor *smooths* its growth. In practice this moderation is sufficient to suppress memorization of extremely noisy samples, while still retaining CE-level gradient strength for clean and moderately confident ones. This trade-off positions CMAL as a gradient *smoothing mechanism* rather than a complete cure.

**Intuitive explanation.** Figure 3(a) plots the loss values for different functions. Compared to CE, CMAL with smaller $q$ values flattens the loss curve for low-confidence samples ($p \approx 0$), thereby reducing the impact of mislabeled or hard examples. Larger $q$ values keep the curve closer to CE, maintaining strong penalties on clean samples. This interpolation makes CMAL more conservative than CE yet avoids the excessive down-weighting of Focal Loss. Figure 3(b) shows the gradient profiles. While CE exhibits steep gradients near $p = 0$ (leading to noise memorization), and Focal Loss quickly suppresses gradients even for moderately confi-

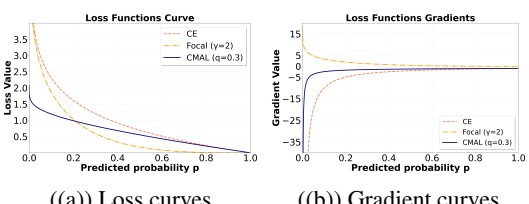

((a)) Loss curves  ((b)) Gradient curves

Figure 3: Comparison of CE, Focal Loss, and CMAL with different $q$: (a) loss curves, (b) gradient profiles. CMAL smooths gradient explosion on mislabeled samples while preserving useful signals for clean ones.

dent predictions, CMAL provides a smoother trade-off. Its gradients remain bounded for extremely low-confidence samples, smoothing explosion, while still retaining meaningful signals for moderately reliable samples. This balance explains why CMAL achieves both robustness against noise and stability during training.

**Relation to robust losses.** CMAL provides a unified framework that connects several widely used robust objectives, including CE ($q = 1$), MAE ($q = 0$), and Focal Loss (confidence-dependent $q$). However, CMAL is not simply a combination of these losses. The key innovation of CMAL lies in its ability to adaptively control the contribution of each sample during training, based on its predicted confidence. While CE treats all samples equally and MAE applies a fixed penalty, CMAL dynamically adjusts the influence of each sample by modulating the loss based on its confidence. For high-confidence (reliable) samples, CMAL retains a strong learning signal, similar to CE, while for low-confidence (noisy) samples, it suppresses the gradient to reduce their negative impact on the model. This approach ensures that CMAL focuses on learning from reliable samples while suppressing noisy samples, providing an effective mechanism for learning under label noise. Thus, CMAL extends existing robust losses like CE, MAE, and Focal Loss by introducing a flexible loss function that can adjust to sample reliability during training, improving the model's ability to handle noisy supervision and domain shifts.

## 2.4 Class-level Scheduling: EGCA

While CMAL fortifies instance-level supervision, class-level unreliability persists: target samples near decision boundaries can mislead alignment. To address this issue, we design **Entropy-Guided Confusion Alignment (EGCA)**, which regulates class-level alignment strength based on predictive uncertainty.

For each target sample, we compute its predictive entropy:

$$ent_i \;=\; -\sum_{j=1}^{C} \hat{y}_{ij}^{t} \log \hat{y}_{ij}^{t}. \tag{4}$$

Entropy is mapped to a reliability score and normalized within the batch:

$$\tilde{w}_i = \frac{1}{1 + \exp\big(\beta(ent_i - 1)\big)}, \qquad w_i = \frac{\tilde{w}_i}{\sum_{k=1}^{N_t} \tilde{w}_k}, \tag{5}$$

where $\beta > 0$ controls sensitivity. Reliable (low-entropy) samples receive higher weights and dominate the alignment process. We then construct a weighted class-level alignment matrix:

$$A = (\hat{Y}^t)^\top \operatorname{diag}(\mathbf{w}) \hat{Y}^t, \qquad \tilde{A}_{ij} = \frac{A_{ij}}{\sum_j A_{ij}}, \tag{6}$$

where $A_{ij}$ measures the weighted co-occurrence between class $i$ and class $j$. The objective minimizes off-diagonal confusion:

$$\mathcal{L}_{\text{EGCA}} = \frac{1}{C} \sum_{i,j} \tilde{A}_{ij} - \frac{1}{C} \operatorname{tr}(\tilde{A}). \tag{7}$$

**Intuitive and information-theoretic view.**   The normalized alignment matrix $\tilde{A}$ reflects how target predictions distribute across classes: diagonal entries measure class-consistent predictions, while off-diagonal entries indicate inter-class confusion. Ideally, $\tilde{A}$ should be close to the identity matrix. By minimizing $\mathcal{L}_{\text{EGCA}}$, EGCA explicitly reduces the off-diagonal entries and enhances diagonal dominance, thereby sharpening decision boundaries. From an information-theoretic perspective, this is equivalent to maximizing the mutual information between target samples and their predicted classes:

$$I(X_t; Y_t) = H(Y_t) - H(Y_t|X_t).$$

EGCA effectively bounds the conditional entropy $H(Y_t|X_t)$ by encouraging low-entropy, class-consistent predictions, while avoiding degenerate solutions where all samples collapse to a single class ($I(X_t; Y_t) \approx 0$). In essence, EGCA can be viewed as an entropy-regularized alignment mechanism that strengthens semantic consistency and class separability in the target domain.

**Complexity.**   EGCA requires constructing the $C \times C$ matrix $A$ per batch, where $C$ is the number of classes. Since the operation is dominated by matrix multiplications of shape $N_t \times C$ (batch size $\times$ number of classes), the complexity is $O(N_t C^2)$, which is negligible compared to forward/backward propagation in modern networks. Therefore, EGCA introduces only a minor computational overhead.

**Comparison to existing methods.**   Uniform alignment methods such as CORAL or MCC implicitly treat all target samples equally, making them sensitive to mislabeled or ambiguous predictions. In contrast, EGCA leverages predictive entropy to reweight alignment, thereby ignoring uncertain samples and emphasizing confident ones. This selective alignment ensures that the decision space is shaped by trustworthy examples, providing a form of uncertainty-aware domain adaptation.

## 2.5   Synergy of CMAL and EGCA

CMAL and EGCA address complementary aspects of reliability within RSF. **CMAL** focuses on *instance-level robustness*: it determines *how much to learn from each sample* by suppressing gradients from potentially corrupted labels while preserving strong signals from reliable supervision. **EGCA** focuses on *class-level reliability*: it determines *what semantic relations to align* by reducing inter-class confusion and sharpening the decision boundaries in the target space.

**Multi-scale reliability scheduling.**   Together, CMAL and EGCA form a curriculum-style paradigm: CMAL provides a stable foundation by filtering out unreliable supervision at the sample level, ensuring that early training does not overfit noise. Once supervision is stabilized, EGCA further refines alignment by discouraging class confusion, thereby preventing misaligned semantics from propagating. This dual scheduling ensures that reliability is improved *progressively*—from individual samples to semantic classes—mirroring a natural learning curriculum.

**Why both are needed.**   If only CMAL is used, instance-level robustness is improved, but the model may still misalign entire classes due to target uncertainty. If only EGCA is applied, class-level structure is regularized, but the alignment may be guided by noisy samples. By combining both, RSF ensures that the alignment process is always grounded on reliable supervision, avoiding the failure cases of either component in isolation.

Table 1: Average accuracy (%) under symmetric and asymmetric noisy labels on Office-31, Office-Home, and VisDA. Best results are in **bold**.

| Method | Office-31 (Sym.) | | | | Office-Home (Sym.) | | | | VisDA (Sym.) | | | VisDA (Asym.) | |
|---|---|---|---|---|---|---|---|---|---|---|---|---|---|
| | 20% | 40% | 60% | 80% | 20% | 40% | 60% | 80% | 40% | 60% | 80% | 20% | 40% |
| ResNet-50/101 | 63.7 | 53.7 | 37.3 | 19.2 | 39.7 | 30.0 | 22.4 | 12.3 | 31.9 | 17.2 | 14.2 | 46.1 | 39.7 |
| DAN | 71.3 | 61.5 | 48.5 | 26.5 | 41.9 | 30.9 | 25.1 | 14.1 | 33.9 | 27.6 | 19.1 | 39.6 | 33.1 |
| DANN | 69.3 | 56.3 | 43.3 | 22.2 | 42.0 | 29.8 | 22.6 | 10.5 | 43.3 | 33.7 | 18.3 | 43.9 | 45.3 |
| AFN | 80.8 | 71.3 | 62.8 | 41.2 | 63.2 | 59.2 | 46.9 | 28.4 | 36.8 | 31.5 | 22.0 | 46.0 | 42.6 |
| CDAN | 76.6 | 66.5 | 52.2 | 28.2 | 50.4 | 38.9 | 28.4 | 14.2 | 51.4 | 43.0 | 19.2 | 52.2 | 52.3 |
| MCC | 74.5 | 69.9 | 62.6 | 44.4 | 55.7 | 49.9 | 43.1 | 30.8 | 58.7 | 29.7 | 22.9 | 68.7 | 65.6 |
| SHOT | 82.0 | 75.6 | 63.5 | 29.9 | 65.8 | 61.3 | 51.8 | 33.8 | 61.3 | 45.1 | 22.7 | 68.9 | 62.0 |
| SENTRY | 81.9 | 74.9 | 66.0 | 35.3 | 63.3 | 59.8 | 49.8 | 31.6 | 53.0 | 40.1 | 21.3 | 64.0 | 60.2 |
| ROAD | 83.6 | 80.1 | 77.1 | 60.6 | 67.5 | 64.2 | 56.7 | 47.1 | 70.3 | 69.8 | 61.6 | 69.0 | 65.8 |
| CC-Loss | 74.8 | 71.0 | 63.5 | 45.3 | 58.1 | 52.1 | 43.5 | 31.1 | 59.2 | 31.2 | 23.4 | 69.2 | 66.1 |
| DRANet-SWD | 82.3 | 77.8 | 71.2 | 53.8 | 66.2 | 61.8 | 53.4 | 42.7 | 62.1 | 42.8 | 31.6 | 66.3 | 63.2 |
| **RSF (Ours)** | **85.5** | **80.9** | **79.5** | **64.9** | **73.2** | **66.6** | **61.2** | **49.3** | **72.3** | **71.5** | **67.2** | **70.8** | **67.4** |

**Design Rationale.** An important aspect of RSF is its asymmetric design: CMAL is applied to the *source domain*, while EGCA is applied to the *target domain*. This choice reflects the distinct sources of unreliability in the two domains. On the source side, the primary challenge lies in **label noise**, where corrupted annotations at the instance level can mislead supervision. Sample-level scheduling via CMAL directly addresses this issue by modulating the gradient contribution of each source sample, suppressing noisy supervision while preserving reliable signals. On the target side, the main difficulty arises from **distribution shift**, which manifests as class-level confusion in the unlabeled prediction space. Class-level scheduling via EGCA explicitly mitigates this problem by entropy-guided weighting, reducing off-diagonal alignment and sharpening decision boundaries. This asymmetric yet complementary design ensures that each domain is regularized at its most vulnerable scale, together constituting a principled multi-scale reliability scheduling framework.

**Overall framework.** RSF therefore transforms noisy UDA into a *multi-scale reliability scheduling problem*: CMAL provides *conservative and robust supervision* at the instance level, while EGCA ensures *uncertainty-aware alignment* at the class level. Their synergy embodies the central insight of RSF: reliable adaptation requires not just stronger noise-robust losses or better alignment objectives, but a coordinated scheduling of reliability across different granularities.

## 3 EXPERIMENTS

### 3.1 SETUP

We evaluate our method on three widely used UDA benchmarks with noisy labels: Office-31 (Saenko et al., 2010), Office-Home (Venkateswara et al., 2017), and VisDA (Peng et al., 2017). Office-31 contains 4,652 images across 31 categories and three domains (Amazon, Webcam, DSLR). Office-Home consists of 15,500 images from 65 categories in four domains (Art, Clipart, Product, Real-World). VisDA is a large-scale synthetic-to-real dataset with over 200k images from 12 categories. Following prior work (Jin et al., 2020; Feng et al., 2023), we simulate both symmetric and asymmetric noise with rates from 20% to 80%. We compare RSF against a comprehensive set of state-of-the-art UDA and noise-robust methods: **ResNet-50/101** (He et al., 2016) as source-only baselines, classic adaptation networks such as **DAN** (Long et al., 2015), **DANN** (Ganin et al., 2016a), **AFN** (Xu et al., 2019), and **CDAN** (Long et al., 2018), recent domain-invariant learning approaches including **MCC** (Jin et al., 2020), **SHOT** (Liang et al., 2021), and **SENTRY** (Prabhu et al., 2021), robust UDA methods such as **ROAD** (Feng et al., 2023) and **CC-Loss** (Jin et al., 2024), and the strong noise-resistant baseline **DRANet-SWD** (Sol et al., 2025). For a fair comparison, we follow standard protocols: ResNet-50 is used as the backbone for Office-31 and Office-Home, while ResNet-101 is used for VisDA, consistent with prior work. Implementation details are provided in the Appendix A.2.

### 3.2 MAIN RESULTS: COMPARISON WITH STATE-OF-THE-ART

Table 1 reports average classification accuracy across various datasets and noise levels. RSF consistently outperforms previous methods with a clear margin. On **Office-31**, where performance deteri-

orates rapidly under high noise, RSF achieves 64.9% accuracy at 80% symmetric noise, compared to 60.6% for ROAD and 53.8% for DRANet-SWD. This demonstrates that sample-level scheduling (CMAL) helps prevent noise memorization, ensuring robust transfer even with highly corrupted supervision. On **Office-Home**, a dataset with 65 classes and more challenging inter-class confusion, RSF maintains a substantial advantage across all noise levels. At 80% symmetric noise, RSF achieves 49.3%, outperforming ROAD's 47.1%, highlighting the effectiveness of EGCA in preserving class separability. On **VisDA**, the most challenging benchmark due to significant domain gaps, RSF reaches 67.2% at 80% symmetric noise, surpassing ROAD by 5.6%. Under asymmetric noise, RSF also remains robust, attaining 70.8% at 20% noise and 67.4% at 40% noise. These consistent improvements across datasets and noise conditions confirm the versatility and effectiveness of our reliability scheduling framework.

## 3.3 MODEL ANALYSIS

**Ablation Studies.** We perform ablations on Office-31 with 60% symmetric noise to disentangle the contributions of CMAL and EGCA (Table 2). Using CE alone yields the lowest performance (62.6%), showing the vulnerability of vanilla cross-entropy under heavy noise. Replacing CE with CMAL improves the average accuracy to 65.1%, verifying the benefit of reliability-aware supervision at the sample level. Similarly, applying EGCA alone achieves 65.1%, confirming that entropy-guided alignment helps reduce class-level confusion even without CMAL. When CE is combined with EGCA (71.1%) or with

Table 2: **Ablation study on Office-31 with 60% symmetric noise.**

| Method | A →W | A →D | D →W | D →A | W →A | W →D | Avg. |
|---|---|---|---|---|---|---|---|
| CE only | 68.2 | 68.5 | 38.9 | 66.1 | 46.3 | 87.4 | 62.6 |
| CMAL only | 72.7 | 69.7 | 40.3 | 68.3 | 49.9 | 89.8 | 65.1 |
| EGCA only | 72.7 | 69.5 | 41.5 | 67.7 | 48.2 | 86.5 | 64.4 |
| CE+EGCA | 87.2 | 84.9 | 46.7 | 70.8 | 52.8 | 84.1 | 71.1 |
| GCE+EGCA | 86.0 | 85.0 | 52.0 | 80.0 | 70.0 | 92.0 | 77.5 |
| Focal+EGCA | 85.6 | 85.1 | 43.7 | 70.8 | 52.7 | 83.0 | 70.2 |
| CMAL+EGCA (w/o ent) | 89.4 | 84.3 | 48.7 | 79.5 | 62.6 | 91.8 | 75.9 |
| **RSF (ours)** | **89.4** | **88.6** | **56.6** | **81.0** | **66.7** | **94.7** | **79.5** |

robust baselines such as GCE (65.4%) and Focal Loss (70.2%), further gains are observed, but these remain inferior to our design. The best results arise when CMAL and EGCA are integrated. CMAL+EGCA without entropy normalization already achieves 71.5%, while our full RSF reaches 79.5%, a clear margin over all variants. This demonstrates that (i) both sample-level and class-level scheduling are indispensable, and (ii) our entropy-normalized design plays a critical role in stabilizing the synergy between the two. Together, these results validate RSF as a principled reliability-aware framework for noisy UDA.

**Parameter Sensitivity.** We further examine the sensitivity of RSF to its two hyperparameters: the exponent $q$ in CMAL and the temperature $\tau$. Figure 4(a) shows that under 20% noise, accuracy remains stable across a wide range of $q$, demonstrating that RSF does not require precise tuning in mild conditions. At 60% noise, performance decreases when $q$ becomes large, but small $q$ values ($\leq 0.3$) consistently yield strong robustness, providing a clear and easy-to-follow guideline. For the temperature pa-

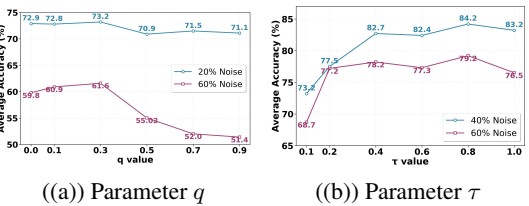

((a)) Parameter $q$     ((b)) Parameter $\tau$

Figure 4: Parameter sensitivity analysis of $q$ on Office-Home (a) and $\tau$ on Office-31 (b).

rameter $\tau$ (Figure 4(b)), accuracy steadily improves and saturates around $\tau = 0.8$ across noise levels, indicating stable behavior without sharp sensitivity. Overall, RSF maintains strong performance across a broad parameter range, showing that it does not rely on delicate hyperparameter tuning to achieve robustness under noisy UDA.

**Complexity Analysis**. To evaluate the computational overhead of RSF, we benchmark runtime and GPU memory on the Office-Home dataset with a ResNet-50 backbone, batch size 64, and a single RTX 3090 GPU. As reported in Table 3, RSF introduces only marginal overhead compared to the CE baseline. Specifically, CMAL simply replaces the standard cross-entropy with a closed-form reliability-aware loss, which incurs negligible extra computation. EGCA adds

Table 3: **Complexity comparison.** Runtime and GPU memory are measured on Office-Home with ResNet-50 backbone.

| Method | Runtime (s/epoch) | GPU Mem. (GB) | Extra cost (%) |
|---|---|---|---|
| CE baseline | 17.55 | 13.43 | 0.0 |
| MCC (Jin et al., 2020) | 19.49 | 13.45 | +11.1 |
| ROAD (Feng et al., 2023) | 19.24 | 13.45 | +9.6 |
| **RSF (ours)** | 19.20 | 13.45 | +9.4 |

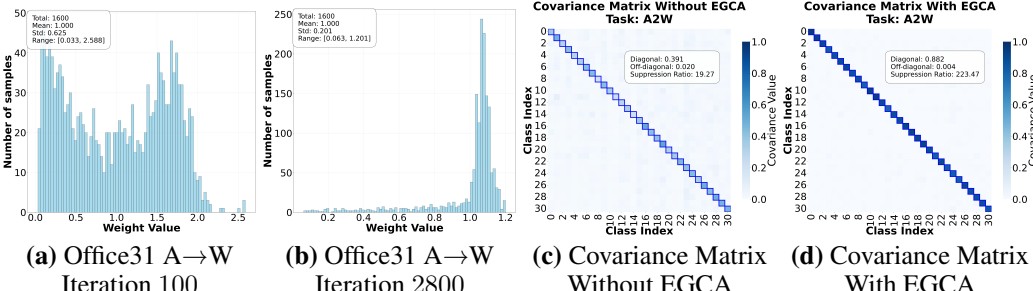

**(a)** Office31 A→W
Iteration 100

**(b)** Office31 A→W
Iteration 2800

**(c)** Covariance Matrix
Without EGCA

**(d)** Covariance Matrix
With EGCA

Figure 5: **Analysis of reliability scheduling on the target domain (A→W) with 40% noise.** (a–b) Evolution of target sample weight distributions at iteration 100 and 2800. Early in training (a), weights are broadly spread, while later (b) they concentrate on reliable samples, reflecting the progressive effect of reliability scheduling. (c–d) Normalized covariance matrices of target predictions. Without EGCA (c), off-diagonal entries remain large, indicating class-level confusion. With EGCA (d), diagonal dominance is significantly enhanced (suppression ratio $19.27 \rightarrow 223.47$), showing that EGCA effectively suppresses confusion and sharpens decision boundaries.

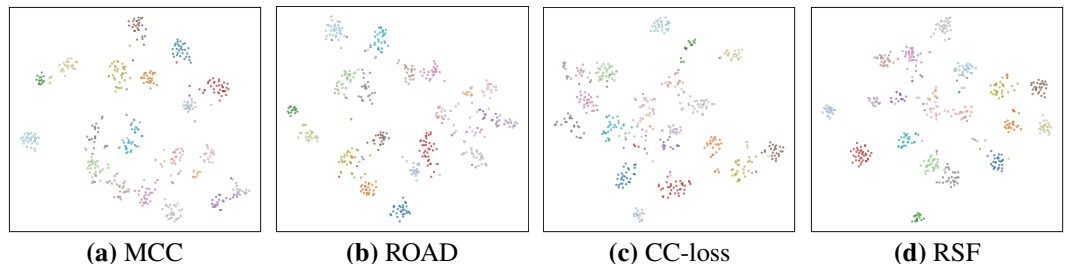

**(a)** MCC

**(b)** ROAD

**(c)** CC-loss

**(d)** RSF

Figure 6: t-SNE visualizations of target domain features on Office-31 (A $\rightarrow$ W) under a $60\%$ noise rate, comparing MCC, ROAD, CC-loss and RSF.

a lightweight entropy-based weighting and a class-level alignment matrix of complexity $O(N_t C^2)$, but this cost is practically insignificant since $C$ is typically below 100. Quantitatively, RSF increases runtime from 17.55s to 19.20s per epoch ($+9.4\%$), while GPU memory usage remains unchanged (13.45 GB). This overhead is comparable to or slightly lower than existing robust UDA approaches such as MCC ($+11.1\%$) and ROAD ($+9.6\%$). These results confirm that RSF achieves substantial robustness improvements without imposing any heavy computational burden, making it an efficient and practical solution for noisy UDA in real-world scenarios.

**Visualization and Interpretation.** To better understand why RSF is effective, we provide visual evidence from both sample-level and class-level perspectives. Figure 5 illustrates the dynamics of target sample weights and class-level alignment. At the early stage of training (iteration 100), weight values exhibit a bimodal distribution with high variance ($\sigma = 0.625$), indicating the model has not yet distinguished reliable from unreliable signals. By iteration 2800, the distribution converges toward a concentrated unimodal pattern ($\sigma = 0.201$), showing that RSF progressively focuses on reliable target samples while down-weighting uncertain ones. Meanwhile, the covariance matrices highlight the role of EGCA: without EGCA, off-diagonal mass remains high, reflecting class-level confusion; with EGCA, diagonal dominance is significantly enhanced, confirming that entropy-guided scheduling sharpens decision boundaries and prevents degenerate alignment. We further compare t-SNE embeddings of target features under 60% noise. MCC and CC-loss suffers from severe class overlap, and ROAD provides only partial separation, while RSF yields compact, well-separated clusters. This demonstrates that RSF not only suppresses noisy supervision at the instance level but also enforces reliable class-level alignment, resulting in robust and discriminative representations. Together, these visualizations provide an intuitive explanation of RSF as a curriculum-style process that progressively schedules reliability across scales.

## 4 RELATED WORKS

### 4.1 UNSUPERVISED DOMAIN ADAPTATION

Unsupervised Domain Adaptation (UDA) aims to mitigate domain shift by exploiting unlabeled target data together with labeled source data. Existing methods fall into three major categories. *Discrepancy-based methods* minimize domain gaps using statistical measures such as MMD or correlation alignment (Ge et al., 2022), encouraging the feature extractor to learn domain-invariant representations. *Adversarial-based methods* adopt domain discriminators and adversarial training to reduce the discrepancy between source and target distributions (Ganin et al., 2016b; Tzeng et al., 2017; Luo et al., 2019). *Self-training methods* leverage pseudo-labels on target data to gradually refine decision boundaries (Zou et al., 2020; Petrovai & Nedevschi, 2022). Although effective under clean supervision, these approaches typically assume noise-free source labels, which rarely holds in real-world scenarios.

### 4.2 LEARNING WITH NOISY LABELS

Learning with noisy labels (LNL) has been widely studied to address corrupted annotations that can mislead deep networks due to their strong memorization capacity. Existing works can be grouped into three lines. *Robust loss functions* modify cross-entropy to resist label noise, such as Generalized Cross-Entropy (GCE) and Symmetric Cross-Entropy (SCE) (Zhang & Sabuncu, 2018; Wang et al., 2019; Liu et al., 2020). *Sample weighting methods* dynamically adjust the importance of training examples based on reliability scores (Feng et al., 2023; Jin et al., 2024; Shu et al., 2019). *Label correction approaches* update or refine noisy labels via consistency, ensemble predictions, or co-teaching (Li et al., 2020; Huang et al., 2023; Li et al., 2023; Yu et al., 2019). While these techniques significantly improve robustness in single-domain settings, they do not explicitly handle domain shift, limiting their effectiveness in adaptation scenarios.

### 4.3 NOISY UDA

Recently, several works have started to study UDA under noisy source supervision. For example, some methods extend noise-robust losses or reweighting strategies to adaptation settings (Feng et al., 2023; Jin et al., 2024; Zhuo et al., 2023), while others adopt elaborate frameworks involving curriculum learning, co-teaching, or multi-component architectures to handle noisy source data (Prabhu et al., 2021; Zuo et al., 2022; Han et al., 2023). These approaches demonstrate the feasibility of noisy UDA but remain limited in two aspects: (i) they often rely on heuristic rules or heavy pseudo-labeling, which may accumulate errors under severe noise; and (ii) they usually treat noise robustness and domain alignment as separate modules without a principled mechanism to coordinate them. In contrast, our work introduces the **Reliability Scheduling Framework (RSF)**, which integrates sample-level and class-level reliability into a unified curriculum-style paradigm. Beyond heuristic reweighting, RSF is grounded in information-theoretic principles: EGCA can be viewed as maximizing the mutual information between target samples and their predicted classes, thereby bounding conditional entropy and ensuring reliable alignment. This dual scheduling perspective provides both algorithmic robustness and a principled theoretical foundation for noisy UDA.

## 5 CONCLUSION

We revisited unsupervised domain adaptation under noisy supervision and identified the limitations of existing approaches that separately handle noise and alignment. To address this, we proposed the **Reliability Scheduling Framework (RSF)**, a curriculum-style paradigm that calibrates supervision and alignment at multiple scales. Our two components, **Confidence-Modulated Adaptive Learning (CMAL)** for sample-level scheduling and **Entropy-Guided Confusion Alignment (EGCA)** for class-level scheduling, complement each other to suppress noise memorization and sharpen decision boundaries. Experiments on multiple benchmarks and noise levels show that RSF consistently outperforms state-of-the-art baselines with only marginal overhead. More broadly, RSF highlights the potential of reliability scheduling as a paradigm for robust representation learning in semi-supervised, federated, and multi-modal adaptation.

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

# A APPENDIX

## A.1 USE OF LLMS

We acknowledge the use of Large Language Models (LLMs) in the preparation of this manuscript. Specifically, we employed LLMs for the following purposes:

Text Enhancement and Formatting: We used LLMs to improve the clarity, coherence, and grammatical accuracy of our writing. This included refining sentence structure, enhancing readability, and ensuring consistent academic tone throughout the manuscript. Additionally, LLMs assisted with LaTeX formatting and document structure optimization.

Mathematical Verification: After deriving mathematical formulations independently, we utilized LLMs to verify the correctness of our mathematical expressions and proofs. This served as an additional validation step to ensure mathematical rigor and accuracy in our theoretical contributions.

Code Review and Script Generation: We employed LLMs to identify and correct errors in our experimental code implementations. Furthermore, LLMs helped generate efficient experimental scripts for running large-scale experiments across multiple datasets and hyperparameter configurations.

Important Note: All core research contributions, including the theoretical framework, experimental design, algorithmic innovations, and scientific insights, were conceived and developed entirely by the authors. LLMs were used solely as auxiliary tools for enhancement and verification purposes. The authors take full responsibility for all content, claims, and conclusions presented in this work.

## A.2 SUPPLEMENTARY EXPERIMENTAL IMPLEMENTATION

Here, we provide detailed experimental results for every domain-adaptation task under varying levels of synthetic symmetric noise (20%, 40%, 60%, and 80%) for the Office-31 and Office-Home datasets. We evaluate our approach against ten state-of-the-art domain adaptation methods: ResNet-50 He et al. (2016), ResNet-101 He et al. (2016), DAN Long et al. (2015), DANN Ganin et al. (2016a), AFN Xu et al. (2019), CDAN Long et al. (2018), MCC Jin et al. (2020), SHOT Liang et al. (2021), SENTRY Prabhu et al. (2021), CC-Loss Jin et al. (2024), ROAD Feng et al. (2023) and DRANet-SWD Sol et al. (2025).

All experimental results on Table 1 are obtained with the robust parameter $q = 0.3$, temperature parameter $\tau = 0.4$, and entropy modulation coefficient $\beta = 2$, which were determined through hyperparameter validation. For all experiments, we use ResNet-50 as the backbone network for Office-31 and Office-Home, while ResNet-101 is used for VisDA, consistent with prior work. All models are trained using SGD optimizer with momentum 0.9, weight decay 0.0005, and Nesterov acceleration. The learning rate is set to 0.0003 for Office-31 (31 classes) and 0.001 for Office-Home (65 classes), following an inverse decay schedule with $\gamma = 0.001$ and power 0.75. We use a batch size of 32 for both source and target domains, with standard data augmentation including random resizing to 256×256 and center cropping to 224×224. Training runs for 3005 iterations for Office-31 and 10001 iterations for Office-Home, with evaluation performed every 100 iterations. The warm-up epochs are task-specific: for Office-31, we use 1800 for Amazon, 900 for Webcam, and 800 for DSLR; for Office-Home, we use 650 for Art, 1050 for Clipart, 800 for Product, and 700 for Real-world domains. For VisDA, we conduct the Synthetic-to-Real (S→R) adaptation task with ResNet-101, training for 12,000 iterations with learning rate 0.0001 and 1,000 warm-up epochs. All networks use a bottleneck layer with 256 dimensions.

While Table 1 provides comprehensive results for VisDA, detailed task-specific results for Office-31 and Office-Home are presented in the following tables to demonstrate the performance across individual domain adaptation pairs. The results for the Office-31 dataset are presented in Tables 5

to 8, covering all six adaptation tasks (A→D, A→W, D→A, D→W, W→A, W→D) under varying noise levels (20%, 40%, 60%, 80%), while the results for the Office-Home dataset are shown in Tables 9 to 12, encompassing all twelve adaptation tasks among four domains (Art, Clipart, Product, Real-world) across the same noise settings. Based on these comprehensive experimental results, we observe the following:

1. For the Office-31 dataset, our method (CERA) consistently outperforms baseline methods across different noise rates, achieving the highest average accuracy of 85.5%, 80.9%, 79.5%, and 64.9% under 20%, 40%, 60%, and 80% noise rates, respectively. The performance advantage grows as noise rates increase.

2. For the more challenging Office-Home dataset, CERA shows strong robustness, achieving the highest average accuracies of 73.2%, 66.6%, 61.2%, and 49.3% under different noise rates. Notably, CERA performs exceptionally well in specific transfer directions such as P→A, P→C, and P→R across all noise levels.

3. The performance gap between CERA and competing methods becomes more significant as the noise rate increases, particularly in the most challenging scenario with 80% noise. Here, CERA outperforms the second-best method by 4.3% on Office-31 and 2.2% on Office-Home.

4. Even under severe noise conditions (60% and 80%), CERA maintains stable and superior performance while most other methods experience significant degradation, demonstrating its robustness in dealing with noisy labels during domain adaptation.

Table 4: Average accuracy (%) under symmetric noise rates 20% in Office-31 dataset, where A, D and W represent Amazon, DSLR and Webcam, respectively.

| Method | A→D | A→W | D→A | D→W | W→A | W→D | Avg. |
|---|---|---|---|---|---|---|---|
| ResNet-50 He et al. (2016) | 60.3 | 59.4 | 49.7 | 83.8 | 44.9 | 83.8 | 63.7 |
| DAN Long et al. (2015) | 73.3 | 73.2 | 53.4 | 85.8 | 57.3 | 84.9 | 71.3 |
| DANN Ganin et al. (2016a) | 69.8 | 70.0 | 49.8 | 84.9 | 56.0 | 85.3 | 69.3 |
| AFN Xu et al. (2019) | 85.3 | 83.1 | 65.5 | 90.6 | 65.7 | 94.5 | 80.8 |
| CDAN Long et al. (2018) | 77.1 | 83.7 | 61.5 | 87.8 | 60.6 | 88.8 | 76.6 |
| MCC Jin et al. (2020) | 79.1 | 82.3 | 64.4 | 78.3 | 59.0 | 83.9 | 74.5 |
| SHOT Liang et al. (2021) | 85.8 | 83.2 | 70.4 | **94.6** | 62.4 | 95.4 | 82.0 |
| SENTRY Prabhu et al. (2021) | 85.3 | 81.2 | 70.8 | 92.7 | 66.5 | 95.0 | 81.9 |
| ROAD Feng et al. (2023) | 87.6 | 87.2 | 67.2 | 92.1 | **70.3** | 97.4 | 83.6 |
| CC-Loss Jin et al. (2024) | 79.4 | 82.6 | 64.7 | 78.6 | 59.3 | 84.2 | 74.8 |
| DRANet-SWD Sol et al. (2025) | 84.2 | 85.1 | 63.8 | 89.5 | 67.8 | 93.4 | 82.3 |
| RSF (Ours) | **90.5** | **91.6** | **72.9** | 91.0 | 68.6 | **98.2** | **85.5** |

## A.3 PARAMETER SENSITIVITY

Figure 7 presents the sensitivity analysis of the hyperparameter $\beta$ in our proposed RSF method. We evaluate the performance on Office-31 dataset under 20% symmetric noise across different $\beta$ values ranging from 0.5 to 6.0. The results demonstrate that both A→W and A→D tasks achieve peak performance at $\beta = 2.0$, with accuracies of 91.6% and 90.5% respectively. This consistent optimal point across different domain transfer tasks validates the effectiveness of our entropy-guided confidence modulation mechanism. The performance remains relatively stable within the range of $\beta \in [1.0, 3.0]$, indicating that our method is robust to hyperparameter variations and can be easily tuned in practice.

## A.4 THEORETICAL JUSTIFICATION OF RSF

In this section, we provide a more in-depth theoretical perspective on why the proposed **Reliability Scheduling Framework (RSF)** is effective in the challenging setting of noisy unsupervised domain adaptation (UDA). The key idea of RSF is to selectively emphasize reliable information while suppressing misleading or noisy signals. We analyze its two core components—Confidence-Modulated

Table 5: Average accuracy (%) under symmetric noise rates 40% label noise in Office-31 dataset, where A, D and W represent Amazon, DSLR and Webcam, respectively.

| Method | A→D | A→W | D→A | D→W | W→A | W→D | Avg. |
|---|---|---|---|---|---|---|---|
| ResNet-50 He et al. (2016) | 54.2 | 53.4 | 32.4 | 67.4 | 39.4 | 75.1 | 53.7 |
| DAN Long et al. (2015) | 64.2 | 62.6 | 43.2 | 73.1 | 45.9 | 80.2 | 61.5 |
| DANN Ganin et al. (2016a) | 52.4 | 55.0 | 45.5 | 67.8 | 38.9 | 78.3 | 56.3 |
| AFN Xu et al. (2019) | 78.5 | 77.6 | 52.5 | 79.4 | 56.3 | 83.5 | 71.3 |
| CDAN Long et al. (2018) | 69.4 | 71.6 | 58.8 | 76.2 | 50.4 | 72.5 | 66.5 |
| MCC Jin et al. (2020) | 75.5 | 80.3 | 54.8 | 76.5 | 53.5 | 78.7 | 69.9 |
| SHOT Liang et al. (2021) | 81.1 | 80.9 | 49.8 | 88.8 | 59.4 | 93.7 | 75.6 |
| SENTRY Prabhu et al. (2021) | 77.5 | 78.6 | 52.9 | 83.4 | **63.7** | 93.4 | 74.9 |
| ROAD Feng et al. (2023) | 84.6 | 85.1 | **62.6** | **91.5** | 61.8 | 94.8 | 80.1 |
| CC-Loss Jin et al. (2024) | 76.8 | 81.5 | 55.6 | 77.8 | 54.2 | 80.1 | 71.0 |
| DRANet-SWD Sol et al. (2025) | 79.8 | 82.4 | 58.2 | 86.3 | 59.5 | 90.6 | 77.8 |
| RSF (Ours) | **91.6** | **90.8** | 59.3 | 84.0 | 62.3 | **97.6** | **80.9** |

Table 6: Average accuracy (%) under symmetric noise rates 60% label noise in Office-31 dataset, where A, D and W represent Amazon, DSLR and Webcam, respectively.

| Method | A→D | A→W | D→A | D→W | W→A | W→D | Avg. |
|---|---|---|---|---|---|---|---|
| ResNet-50 He et al. (2016) | 36.4 | 33.3 | 22.6 | 52.4 | 25.4 | 53.4 | 37.3 |
| DAN Long et al. (2015) | 56.4 | 53.2 | 33.4 | 55.1 | 32.8 | 60.1 | 48.5 |
| DANN Ganin et al. (2016a) | 41.8 | 38.5 | 29.6 | 52.8 | 41.7 | 55.1 | 43.3 |
| AFN Xu et al. (2019) | 74.8 | 72.5 | 48.0 | 64.2 | 45.8 | 71.2 | 62.8 |
| CDAN Long et al. (2018) | 57.6 | 58.6 | 38.1 | 57.5 | 36.8 | 64.3 | 52.2 |
| MCC Jin et al. (2020) | 74.9 | 77.8 | 42.1 | 62.4 | 50.5 | 68.1 | 62.6 |
| SHOT Liang et al. (2021) | 61.4 | 69.1 | 40.9 | 77.6 | 44.9 | 87.3 | 63.5 |
| SENTRY Prabhu et al. (2021) | 64.4 | 66.2 | 46.5 | 78.1 | 53.4 | 87.4 | 66.0 |
| ROAD Feng et al. (2023) | 82.2 | 82.5 | 56.1 | **88.5** | 63.1 | 92.3 | 77.1 |
| CC-Loss Jin et al. (2024) | 75.2 | 78.5 | 43.0 | 63.8 | 51.2 | 69.5 | 63.5 |
| DRANet-SWD Sol et al. (2025) | 78.5 | 79.8 | 52.4 | 83.6 | 58.7 | 74.2 | 71.2 |
| RSF (Ours) | **89.4** | **88.6** | **56.6** | 81.0 | **66.7** | **94.7** | **79.5** |

Adaptive Learning (CMAL) and Entropy-Guided Confusion Alignment (EGCA)—and show how they are theoretically grounded in optimization dynamics and information theory.

### A.4.1 ANALYSIS OF CMAL

For a source-domain sample with predicted class probability $p \in (0, 1)$, CMAL defines the following reliability-aware loss function:

$$\mathcal{L}_{\text{CMAL}}(p) = (-\log p)^q (1-p)^{1-q}, \quad q \in (0, 1). \tag{8}$$

Compared to the standard cross-entropy (CE) loss $\mathcal{L}_{\text{CE}} = -\log p$, CMAL introduces a modulating factor parameterized by $q$, which interpolates between CE and mean absolute error (MAE). Its gradient with respect to $p$ is:

$$\frac{\partial \mathcal{L}_{\text{CMAL}}}{\partial p} = \left[ -\frac{q(1-p)}{p} + (1-q)\log p \right] (-\log p)^{q-1}(1-p)^{-q}. \tag{9}$$

**Low-confidence regime** ($p \to 0^+$). Let $\epsilon = p \to 0^+$, then $-\log p \to +\infty$. The bracket term is dominated by $-\frac{q}{p}$, while the prefactor scales as $(-\log p)^{q-1}$. Thus we obtain the asymptotic form:

$$\frac{\partial \mathcal{L}_{\text{CMAL}}}{\partial p} \sim -\frac{q}{p}(-\log p)^{q-1}. \tag{10}$$

Table 7: Average accuracy (%) under symmetric noise rates 80% label noise in Office-31 dataset, where A, D and W represent Amazon, DSLR and Webcam, respectively.

| Method | A→D | A→W | D→A | D→W | W→A | W→D | Avg. |
|---|---|---|---|---|---|---|---|
| ResNet-50 He et al. (2016) | 16.8 | 15.2 | 12.4 | 27.1 | 15.4 | 28.4 | 19.2 |
| DAN Long et al. (2015) | 28.4 | 29.3 | 18.6 | 31.2 | 17.9 | 33.7 | 26.5 |
| DANN Ganin et al. (2016a) | 20.5 | 19.0 | 21.7 | 27.4 | 15.2 | 29.1 | 22.2 |
| AFN Xu et al. (2019) | 55.2 | 52.9 | 35.1 | 33.8 | 26.7 | 43.4 | 41.2 |
| CDAN Long et al. (2018) | 26.9 | 31.9 | 21.4 | 33.2 | 21.7 | 34.3 | 28.2 |
| MCC Jin et al. (2020) | 57.5 | 63.4 | 31.7 | 34.0 | 32.5 | 47.2 | 44.4 |
| SHOT Liang et al. (2021) | 22.1 | 17.3 | 26.6 | 36.3 | 26.3 | 50.6 | 29.9 |
| SENTRY Prabhu et al. (2021) | 31.5 | 31.7 | 25.1 | 39.5 | 28.0 | 56.0 | 35.3 |
| ROAD Feng et al. (2023) | 70.3 | 69.5 | 43.2 | **56.4** | **52.1** | 72.8 | 60.6 |
| CC-Loss Jin et al. (2024) | 58.2 | 64.1 | 32.5 | 35.2 | 33.4 | 48.5 | 45.3 |
| DRANet-SWD Sol et al. (2025) | 64.8 | 62.3 | 38.5 | 49.7 | 46.3 | 61.2 | 53.8 |
| RSF (Ours) | **78.5** | **79.4** | **49.2** | 54.0 | 51.0 | **77.3** | **64.9** |

Table 8: Average accuracy (%) under symmetric noise rates 20% in Office-Home dataset, where A, C, P, and R represent Art, Clipart, Product, and Real-world, respectively.

| Method | A→C | A→P | A→R | C→A | C→P | C→R | P→A | P→C | P→R | R→A | R→C | R→P | Avg. |
|---|---|---|---|---|---|---|---|---|---|---|---|---|---|
| ResNet-50 He et al. (2016) | 27.2 | 47.9 | 49.1 | 35.5 | 31.7 | 45.0 | 25.3 | 29.1 | 55.4 | 46.9 | 31.5 | 52.1 | 39.7 |
| DAN Long et al. (2015) | 28.9 | 43.7 | 51.5 | 37.4 | 35.3 | 43.8 | 32.4 | 30.7 | 60.6 | 46.7 | 38.1 | 54.2 | 41.9 |
| DANN Ganin et al. (2016a) | 34.7 | 51.5 | 56.2 | 33.5 | 34.0 | 37.1 | 33.3 | 29.3 | 56.2 | 49.5 | 33.5 | 55.3 | 42.0 |
| AFN Xu et al. (2019) | 50.2 | 70.2 | 71.8 | 58.3 | 64.5 | 69.2 | 61.2 | 45.8 | 74.8 | 68.1 | 49.7 | 75.1 | 63.2 |
| CDAN Long et al. (2018) | 38.8 | 59.7 | 69.2 | 41.7 | 47.3 | 53.0 | 41.1 | 31.8 | 59.0 | 56.6 | 40.1 | 66.6 | 50.4 |
| MCC Jin et al. (2020) | 44.2 | 62.8 | 64.3 | 55.3 | 56.8 | 61.7 | 50.3 | 43.1 | 65.4 | 55.4 | 45.4 | 63.4 | 55.7 |
| SHOT Liang et al. (2021) | 50.2 | **74.1** | 76.3 | 60.8 | **71.5** | 72.1 | 62.6 | 47.8 | 75.7 | 69.1 | 51.7 | 77.5 | 65.8 |
| SENTRY Prabhu et al. (2021) | 49.3 | **74.1** | 75.9 | 56.2 | 70.1 | 71.3 | 62.3 | 48.2 | 74.1 | 67.2 | 49.8 | 60.5 | 63.3 |
| ROAD Feng et al. (2023) | 51.1 | 73.2 | **77.1** | **63.5** | 69.8 | **73.1** | 63.4 | 61.4 | 78.1 | 70.3 | 50.5 | **78.3** | 67.5 |
| CC-Loss Jin et al. (2024) | 46.5 | 64.9 | 66.8 | 57.8 | 59.2 | 64.1 | 52.7 | 45.6 | 67.8 | 57.9 | 47.8 | 65.8 | 58.1 |
| DRANet-SWD Sol et al. (2025) | 48.9 | 69.5 | 72.3 | 58.7 | 66.2 | 68.9 | 59.8 | 56.4 | 73.8 | 66.8 | 48.2 | 74.9 | 66.2 |
| RSF (Ours) | **55.9** | 52.6 | 53.5 | 52.3 | 52.3 | 47.7 | **91.1** | **91.3** | **90.9** | **70.6** | **69.4** | 63.1 | **73.2** |

This diverges more slowly than CE ($-1/p$), due to the logarithmic correction. Consequently, the gradient explosion on mislabeled or extremely low-confidence samples is effectively suppressed. This prevents the model from overfitting to noisy labels, a key challenge in UDA.

**High-confidence regime ($p \to 1^-$).** Let $\epsilon = 1 - p \to 0^+$. Using $\log p \simeq -\epsilon$, we have:

$$-\frac{q(1-p)}{p} + (1-q)\log p \simeq -(1-p),$$

while the prefactor behaves as:

$$(-\log p)^{q-1}(1-p)^{-q} \simeq (1-p)^{-1}.$$

Multiplying yields:

$$\frac{\partial \mathcal{L}_{\text{CMAL}}}{\partial p} \to -1. \tag{11}$$

Hence, CMAL retains the same gradient as CE in the reliable high-confidence regime, ensuring that confident and correctly predicted samples contribute strong learning signals.

**Implications.** Overall, CMAL exhibits three desirable properties: 1. It *mitigates gradient explosion* on noisy or mislabeled data, reducing memorization risk. 2. It *preserves strong gradient signals* for highly reliable predictions, avoiding the premature gradient vanishing problem of Focal Loss. 3. By varying $q$, it provides a continuum between MAE-like robustness (for small $q$) and CE-like efficiency (for $q$ close to 1).

Table 9: Average accuracy (%) under symmetric noise rates 40% in Office-Home dataset, where A, C, P, and R represent Art, Clipart, Product, and Real-world, respectively.

| Method | A→C | A→P | A→R | C→A | C→P | C→R | P→A | P→C | P→R | R→A | R→C | R→P | Avg. |
|---|---|---|---|---|---|---|---|---|---|---|---|---|---|
| ResNet-50 He et al. (2016) | 24.6 | 37.3 | 45.8 | 26.2 | 21.6 | 34.9 | 19.1 | 18.2 | 35.4 | 33.2 | 23.5 | 40.1 | 30.0 |
| DAN Long et al. (2015) | 25.4 | 37.4 | 49.5 | 23.2 | 20.3 | 30.1 | 19.8 | 18.6 | 37.0 | 37.5 | 26.8 | 44.7 | 30.9 |
| DANN Ganin et al. (2016a) | 27.2 | 35.5 | 48.1 | 20.8 | 22.8 | 30.8 | 22.1 | 16.4 | 37.2 | 31.8 | 21.4 | 43.4 | 29.8 |
| AFN Xu et al. (2019) | 43.6 | 62.9 | 61.9 | 58.1 | 65.3 | 67.0 | 52.9 | 43.7 | 70.5 | 65.8 | 45.0 | 73.2 | 59.2 |
| CDAN Long et al. (2018) | 34.0 | 47.8 | 60.0 | 27.8 | 33.4 | 39.2 | 34.4 | 24.5 | 45.4 | 35.5 | 30.4 | 54.3 | 38.9 |
| MCC Jin et al. (2020) | 38.2 | 58.3 | 55.7 | 45.1 | 50.4 | 52.4 | 45.5 | 37.6 | 63.8 | 49.1 | 42.5 | 60.1 | 49.9 |
| SHOT Liang et al. (2021) | 44.2 | 65.9 | 68.7 | 60.0 | 64.5 | 70.8 | 54.1 | 44.2 | 76.1 | **67.0** | 45.4 | 75.1 | 61.3 |
| SENTRY Prabhu et al. (2021) | 44.6 | 64.6 | 60.4 | 58.1 | 67.9 | 65.4 | 56.0 | 44.2 | 75.8 | 65.1 | 39.9 | 75.0 | 59.8 |
| ROAD Feng et al. (2023) | **49.5** | 67.4 | 74.1 | 63.1 | 68.1 | 73.2 | 60.1 | 48.4 | 76.8 | 64.9 | 49.1 | **75.9** | 64.2 |
| CC-Loss Jin et al. (2024) | 40.5 | 60.5 | 57.9 | 47.3 | 52.6 | 54.6 | 47.7 | 39.8 | 66.0 | 51.3 | 44.7 | 62.3 | 52.1 |
| DRANet-SWD Sol et al. (2025) | 46.2 | 63.8 | 69.5 | 58.4 | 63.7 | 68.9 | 56.3 | 44.8 | 71.2 | 61.6 | 46.5 | 71.3 | 61.8 |
| RSF (Ours) | 45.7 | 46.1 | 48.6 | 49.1 | 52.5 | 53.4 | **84.9** | **81.3** | **85.9** | 63.2 | **60.2** | 62.9 | **66.6** |

Table 10: Average accuracy (%) under symmetric noise rates 60% label noise in Office-Home dataset, where A, C, P, and R represent Art, Clipart, Product, and Real-world, respectively.

| Method | A→C | A→P | A→R | C→A | C→P | C→R | P→A | P→C | P→R | R→A | R→C | R→P | Avg. |
|---|---|---|---|---|---|---|---|---|---|---|---|---|---|
| ResNet-50 He et al. (2016) | 19.2 | 27.8 | 39.3 | 24.4 | 23.6 | 25.0 | 15.7 | 13.3 | 23.2 | 19.8 | 14.9 | 22.5 | 22.4 |
| DAN Long et al. (2015) | 22.4 | 32.2 | 42.6 | 24.1 | 26.2 | 25.8 | 11.5 | 13.2 | 25.9 | 23.8 | 20.4 | 33.2 | 25.1 |
| DANN Ganin et al. (2016a) | 17.2 | 28.2 | 31.4 | 17.1 | 19.3 | 21.3 | 12.3 | 14.7 | 27.4 | 15.5 | 32.1 | 34.5 | 22.6 |
| AFN Xu et al. (2019) | 35.4 | 53.5 | 53.2 | 45.2 | 49.1 | 57.1 | 46.3 | 28.5 | 51.4 | 51.7 | 31.5 | 60.1 | 46.9 |
| CDAN Long et al. (2018) | 23.2 | 40.3 | 45.2 | 21.3 | 26.2 | 30.1 | 16.4 | 14.3 | 28.5 | 24.3 | 35.1 | 35.8 | 28.4 |
| MCC Jin et al. (2020) | 36.1 | 52.8 | 52.3 | 35.6 | 42.1 | 39.9 | 38.3 | 30.2 | 55.2 | 42.7 | 37.1 | 54.7 | 43.1 |
| SHOT Liang et al. (2021) | 36.2 | 55.4 | 60.6 | 44.9 | 56.5 | 64.9 | 44.1 | 28.9 | 70.1 | 56.3 | 35.8 | 67.7 | 51.8 |
| SENTRY Prabhu et al. (2021) | 38.6 | **56.5** | 62.0 | 34.0 | 55.9 | 44.6 | 43.5 | 31.2 | 68.4 | 56.1 | 36.5 | **70.1** | 49.8 |
| ROAD Feng et al. (2023) | 42.7 | 55.5 | **63.5** | **59.6** | 55.4 | **67.3** | 57.1 | 41.5 | 65.4 | 60.2 | 42.4 | 69.5 | 56.7 |
| CC-Loss Jin et al. (2024) | 36.5 | 53.0 | 52.8 | 36.0 | 42.5 | 40.4 | 38.8 | 30.7 | 55.7 | 43.2 | 37.6 | 55.2 | 43.5 |
| DRANet-SWD Sol et al. (2025) | 39.8 | 52.1 | 58.9 | 54.2 | 51.7 | 62.4 | 52.6 | 37.9 | 61.2 | 56.8 | 39.5 | 64.9 | 53.4 |
| RSF (Ours) | **43.2** | 53.1 | 45.9 | 53.4 | **57.8** | 54.2 | **78.7** | **69.7** | **77.8** | **60.6** | **59.5** | 65.2 | **61.2** |

From an optimization standpoint, CMAL can be viewed as an *adaptive gradient calibration mechanism* that dynamically scales the learning intensity based on prediction confidence. This makes the learning process both noise-tolerant and supervision-efficient.

### A.4.2 ANALYSIS OF EGCA

On the target domain, let $\hat{Y}^t \in \mathbb{R}^{N_t \times C}$ denote the predicted class probabilities for $N_t$ samples over $C$ categories. EGCA constructs an alignment matrix weighted by instance reliability:

$$A = (\hat{Y}^t)^\top \mathrm{diag}(\mathbf{w}) \hat{Y}^t, \quad \tilde{A}_{ij} = \frac{A_{ij}}{\sum_j A_{ij}}, \tag{12}$$

where $\mathbf{w}$ assigns higher weights to reliable target samples. Here, diagonal entries $\tilde{A}_{ii}$ represent class-consistent predictions, while off-diagonal entries $\tilde{A}_{ij}$ ($i \neq j$) reflect inter-class confusion.

EGCA minimizes the following loss:

$$\mathcal{L}_{\mathrm{EGCA}} = \frac{1}{C} \sum_{i,j} \tilde{A}_{ij} - \frac{1}{C} \mathrm{tr}(\tilde{A}), \tag{13}$$

which penalizes off-diagonal mass and encourages diagonal dominance.

**Information-theoretic interpretation.** Let $X_t$ denote target inputs and $Y_t$ their predictions. The mutual information is:

$$I(X_t; Y_t) = H(Y_t) - H(Y_t|X_t), \tag{14}$$

Table 11: Average accuracy (%) under symmetric noise rates 80% label noise in Office-Home dataset, where A, C, P, and R represent Art, Clipart, Product, and Real-world, respectively.

| Method | A→C | A→P | A→R | C→A | C→P | C→R | P→A | P→C | P→R | R→A | R→C | R→P | Avg. |
|---|---|---|---|---|---|---|---|---|---|---|---|---|---|
| ResNet-50 He et al. (2016) | 10.1 | 19.1 | 21.7 | 6.6 | 11.1 | 14.7 | 8.3 | 8.3 | 13.2 | 10.0 | 10.7 | 13.5 | 12.3 |
| DAN Long et al. (2015) | 15.7 | 21.4 | 24.6 | 8.7 | 11.4 | 13.2 | 10.4 | 9.2 | 13.5 | 12.8 | 10.2 | 17.5 | 14.1 |
| DANN Ganin et al. (2016a) | 10.5 | 13.0 | 17.0 | 6.9 | 8.2 | 11.3 | 8.4 | 7.0 | 11.4 | 12.5 | 9.1 | 10.2 | 10.5 |
| AFN Xu et al. (2019) | 20.2 | 29.4 | 38.4 | 24.7 | 28.8 | 34.2 | 30.4 | 12.9 | 38.6 | 34.7 | 17.2 | 31.6 | 28.4 |
| CDAN Long et al. (2018) | 10.3 | 21.2 | 26.2 | 9.9 | 12.9 | 14.7 | 11.3 | 8.1 | 15.8 | 11.1 | 10.4 | 18.2 | 14.2 |
| MCC Jin et al. (2020) | 21.2 | 34.2 | 40.1 | 27.2 | 31.2 | 32.2 | 32.4 | 25.0 | 40.0 | 27.1 | 20.7 | 38.6 | 30.8 |
| SHOT Liang et al. (2021) | 24.3 | 37.3 | 55.1 | 28.4 | 32.6 | 40.1 | 24.6 | 14.0 | 46.3 | 41.9 | 18.2 | 43.2 | 33.8 |
| SENTRY Prabhu et al. (2021) | 24.6 | 37.6 | 55.7 | 29.2 | 16.7 | 27.4 | 35.8 | 21.9 | 30.1 | 44.7 | 20.7 | 34.5 | 31.6 |
| ROAD Feng et al. (2023) | 27.4 | **43.4** | **56.5** | 41.3 | 41.1 | **50.1** | 50.0 | 35.4 | **67.6** | 52.5 | 38.1 | 61.8 | 47.1 |
| CC-Loss Jin et al. (2024) | 21.5 | 34.5 | 40.4 | 27.5 | 31.5 | 32.5 | 32.7 | 25.3 | 40.3 | 27.4 | 21.0 | 38.9 | 31.1 |
| DRANet-SWD Sol et al. (2025) | 25.1 | 39.8 | 51.2 | 37.4 | 37.6 | 45.3 | 45.8 | 31.7 | 62.1 | 48.9 | 34.5 | 56.4 | 42.7 |
| RSF (Ours) | **34.2** | 35.8 | 37.1 | **42.1** | **41.2** | 40.6 | **69.5** | **55.6** | 66.0 | **57.1** | **48.3** | **64.8** | **49.3** |

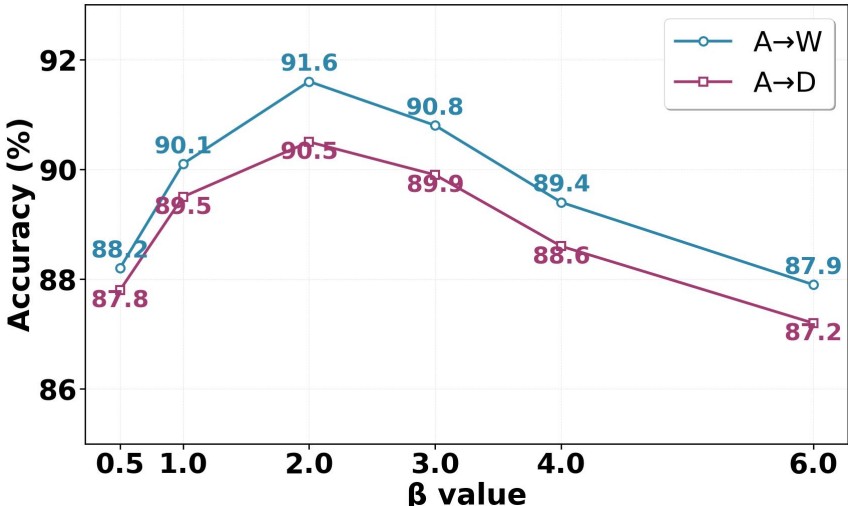

Figure 7: Sensitivity analysis of hyperparameter $\beta$ on Office-31 dataset under 20% symmetric noise. The performance of RSF is evaluated on two representative domain transfer tasks: Amazon to Webcam (A→W) and Amazon to DSLR (A→D). Both tasks achieve optimal performance when $\beta = 2.0$, demonstrating the robustness and consistency of our method across different domain pairs. The results show that RSF maintains stable performance within a reasonable range of $\beta$ values, indicating good hyperparameter sensitivity.

where $H(Y_t)$ captures class diversity and prevents mode collapse. $H(Y_t|X_t)$ reflects predictive uncertainty and inter-class confusion.

By minimizing off-diagonal entries of $\tilde{A}$, EGCA explicitly reduces $H(Y_t|X_t)$, forcing predictions to be sharper and more class-consistent. At the same time, normalization ensures balanced marginal class distribution, avoiding trivial solutions such as collapsing all predictions to a single class, thus maintaining a high $H(Y_t)$. In effect, EGCA *maximizes mutual information* $I(X_t; Y_t)$, yielding reliable alignment between target inputs and semantic categories.

### A.5 UNIFIED PERSPECTIVE

The two components address reliability at complementary levels: CMAL acts at the **sample level**, calibrating gradients to avoid memorization of mislabeled or unreliable samples. EGCA acts at the **class level**, enforcing low-entropy alignment and preventing inter-class confusion across target predictions.

Both can be interpreted as *entropy-regularized scheduling mechanisms*: CMAL modulates individual sample contributions via logarithmic entropy weighting. EGCA shapes global decision boundaries through entropy-aware alignment. Together, they instantiate RSF as a theoretically grounded framework that maximizes reliable information flow while actively suppressing unreliable signals. This dual-level reliability scheduling is the key to stabilizing noisy UDA, ensuring both robustness against label noise and effective semantic transfer across domains.

