# OpenReview forum: "Reliability Scheduling for Robust Domain Adaptation under Label Noise"
_ICLR.cc/2026/Conference — ICLR 2026 Conference Withdrawn Submission_

### Official Review · Reviewer_KJQS · 2025-10-23

**Soundness:** 3
**Presentation:** 2
**Contribution:** 3
**Rating:** 4
**Confidence:** 4

**Summary:**

This paper proposes a new framework called the Reliability Scheduling Framework (RSF) to address unsupervised domain adaptation (UDA) when source labels are noisy. RSF unifies noisy-label learning and domain adaptation.

The poposed methodology introduces a curriculum-style framework which relies on two complementary  scheduling mechanisms:

1. Confidence-Modulated Adaptive Learning (CMAL), which works at sample level by controlling the amount of supervision (in order to mitigate noise overfitting) through a reliability-aware loss in place of standard cross-entropy

2. Entropy-Guided Confusion Alignment (EGCA) which works at class level by reweighing alignment between source and target domains using prediction entropy: high-confidence (low-entropy) target predictions get more weight.

Evaluations are conducted on Office-31, Office-Home, and VisDA under both symmetric and asymmetric noise where RSF consistently outperforms all baselines (MCC, SHOT, SENTRY, ROAD, CC-Loss, DRANet-SWD). Gains are largest under high noise.

The method has a low (~10%) computational overhead.

Ablation show how the two components work in combination, alone, and combined with other methods.

**Strengths:**

The two proposed mechanisms — Confidence-Modulated Adaptive Learning (CMAL) and Entropy-Guided Confusion Alignment (EGCA), are both technically sound and intuitively clear.

CMAL generalizes and improves upon existing robust losses (CE, MAE, Focal Loss) by adaptively modulating gradients based on confidence.

EGCA introduces an entropy-weighted class alignment scheme grounded in information theory, explicitly reducing inter-class confusion while maintaining discriminability.

I have appreciated the gradient analysis, which theoretically justified CMAL’s stability, and hthe information-theoretic interpretation of EGCA in terms of mutual information maximization.

**Weaknesses:**

I think the paper could be improved with a more thorough experimental validation, although I find it sufficient.

Here follow some points of discussion:

1. The experimental results are strong but limited to visual domain adaptation benchmarks (Office-31, Office-Home, VisDA). The framework’s generality would be more convincing if tested on non-visual or cross-modal UDA tasks or real-world noisy datasets.

2. The symmetric and asymmetric noise utilized are not very realistic and it has been shown (e.g. in [1]) that in those cases noisy samples can be easily separated (to be treated/weighted differently). One more realistic source of noise is domain shift itself.

3. No comparison under different backbone architectures (e.g., Vision Transformers, ConvNeXt) — all results rely on ResNet-50,/101, which may not reflect modern architectures.

4. There is a lot of literature addressing the problem of source-free adaptation, where the problem reduces to
   - training on source
   - run inference on the target (producing noisy pseudo-labels)
   - de-noise the target labels / robust learning on the noisy pseudo-labelled target set
   It would be interesting to see how CMAL alone would work in this settings

5. In appendix A.2 the authors mention "hyperparameter validation". Validation is a long-standing issue in UDA. In principle one should not peek at target metrics for tuning the hyper-parameters, since this would mean validating on the test set. On the other hand, source metrics are not informative. One strategy is to use a toy dataset for validation of the hyper-parameters (e.g. SVHN to MNIST) and then use the same hyper-parameters for the other benchmarks. Alternatively, [2] proposes a criterion based on the estimation of the entropy on the predictions of the target. The paper is not discussing the validation issue in any respect.

[1] Cleaning Noisy Labels by Negative Ensemble Learning for Source-Free Unsupervised Domain Adaptation - WACV 2022

[2] Minimal-Entropy Correlation Alignment for Unsupervised Deep Domain Adaptation - ICLR 2018

**Questions:**

I think the paper only needs a slightly stronger experimental evaluation. In particular, see my point n. 1 and 3.

Point 2 is easy to address especially in multi-domain datasets such as Office: instead of manually injecting symmetric or asymmetric noise into source label, one could use a model trained on another domain (not the source, not the target) in order to infer noisy pseudo-labels for the source domain, to be used a noisy ground-truth. This would represent a more realistic and challenging noise.

Point 4 is more of a curiosity.

Point 5 is actually quite critical since many UDA works are indeed validating on the target set in practice.

---

### Official Review · Reviewer_Qzwd · 2025-10-27

**Soundness:** 3
**Presentation:** 3
**Contribution:** 2
**Rating:** 6
**Confidence:** 4

**Summary:**

This paper addresses the problem of Unsupervised Domain Adaptation (UDA) under label noise and proposes a Reliability Scheduling Framework (RSF), which decomposes the problem into two reliability scheduling mechanisms at the sample level and class level. The sample-level module, Confidence-Modulated Adaptive Learning (CMAL), suppresses the impact of noisy samples by modulating gradients based on confidence. The class-level module, Entropy-Guided Confusion Alignment (EGCA), adjusts the alignment strength using predicted entropy to reduce inter-class confusion. Experiments validate the framework’s robustness in high-noise environments on benchmark datasets such as Office-31, Office-Home, and VisDA. However, there are issues regarding the innovativeness of RSF and certain details of the paper.

**Strengths:**

- The paper focuses on unsupervised domain adaptation under label noise and proposes a Reliability Scheduling Framework (RSF) that decomposes the problem into sample-level and class-level reliability mechanisms with a clear structure.
- Gradient visualization (Fig. 3b) and ablation studies (Table 2) demonstrate that the proposed CMAL effectively stabilizes training and improves robustness under noisy supervision.
- Extensive experiments on standard benchmarks show that RSF achieves superior robustness, outperforming SOTA methods under high noise levels (60% and 80%).
- Moreover, RSF is a parameter-efficient approach.

**Weaknesses:**

The strategy of CMAL to enhance learning for high-confidence samples and suppress learning for low-confidence samples is not novel in recent fields of UDA, SF-UDA  or even TTA. For instance: [1] applies weighting to pseudo-label losses; [2] performs consistency-based filtering of pseudo-labels; [3] introduces a "confusion metric" to distinguish sample reliability; [4] partitions data into high-confidence subsets and uncertain subsets, and strengthens learning on the former. Essentially, these methods all filter samples based on confidence, and their core idea is nearly consistent with that of CMAL in this paper.

[1] Guiding Pseudo-Labels with Uncertainty Estimation for Source-free Unsupervised Domain Adaptation, CVPR'23
[2] ProtoCon: Pseudo-Label Refinement via Online Clustering and Prototypical Consistency, CVPR'23
[3] De-Confusing Pseudo-Labels in Source-Free Domain Adaptation, ECCV'24
[4] Selective Label Enhancement Learning for Test-Time Adaptation, ICLR'25

**Questions:**

There are several questions that need to be clarified, as they may affect the final rating:
1. The core mechanism of CMAL appears conceptually similar to several existing methods such as confidence-based filtering or entropy-weighted pseudo-labeling.Could the authors elaborate on the specific advantages or unique design aspects of CMAL compared to these approaches? For instance, does it introduce any new theoretical justification or performance improvement in terms of gradient modulation, dynamic weighting, or stability?
2. The paper proposes EGCA, which reduces inter-class confusion through an entropy-weighted alignment matrix at the class level.
However, prior works [2] and [3] also address inter-class confusion via prototypical consistency constraints and confusion-matrix regularization, respectively. Could the authors further explain how EGCA differs from or improves upon these methods, e.g., in terms of adaptivity, information utilization, or computational efficiency?
3. Since RSF integrates CMAL and EGCA—both of which build upon established ideas—the strong performance observed under high noise conditions is somewhat unexpected. It would strengthen the paper if the authors could provide additional visualizations, comparative experiments, or pseudocode to illustrate how the proposed framework achieves its advantages and to make its effectiveness more convincing.

---

### Official Review · Reviewer_AdBf · 2025-10-31

**Soundness:** 2
**Presentation:** 3
**Contribution:** 3
**Rating:** 4
**Confidence:** 5

**Summary:**

Unsupervised Domain Adaptation (UDA) becomes particularly challenging when source labels are noisy, as label noise and domain shift together cause reliability problems: corrupted labels mislead sample-level supervision, while ambiguous predictions hinder class-level alignment. Existing methods usually tackle these two issues in isolation with static heuristics, resulting in fragile adaptation when noise is severe. The proposed Reliability Scheduling Framework (RSF) unifies noisy-label learning and domain adaptation via multi-scale reliability scheduling: Confidence-Modulated Adaptive Learning (CMAL) adjusts gradients dynamically to suppress noise memorization at the sample level, and Entropy-Guided Confusion Alignment (EGCA) reweights alignment to sharpen decision boundaries at the class level. Experiments on Office-31, Office-Home, and VisDA demonstrate that RSF consistently outperforms state-of-the-art methods under both symmetric and asymmetric noise, proving it an effective solution for robust UDA with noisy supervision.

**Strengths:**

1. The paper is easy to follow.

2. According to the provided experiments, the proposed technique is effective.

3. It is good to see the comparision between Focal loss and the proposed CMAL.

4. The author provide various experiments from multiple perspectives like ComplexityAnalysis and Visualization.

**Weaknesses:**

1. This field has plenty of works like [1] [2] [3] which should be compared in experiments. The authors discuss these method in related work, and mention that "they usually treat noise  robustness and domain alignment as separate modules without aprincipled mechanism to coordinate them. Incontrast,our work introduces the Reliability Scheduling Framework (RSF), which  integrates sample-level and class-level reliability into aunified curriculum-style paradigm." Tha authors should provide experimental evidence to support their claim.

[1] Z. Han, X. Gui, C. Cui, and Y. Yin, “Towards accurate and robust domain adaptation under noisy environments,” in Proc. 29th Int. Joint Conf. Artif. Intell., C. Bessiere, Ed., 2020, pp. 2269–2276.
[2] Y. Zuo, H. Yao, L. Zhuang, and C. Xu, “Seek common ground while reserving differences: A model-agnostic module for noisy domain adaptation,” IEEE Trans. Multimedia, vol. 24, pp. 1020–1030, 2022.
[3] Junbao Zhuo, Shuhui Wang, Qingming Huang. Uncertainty modeling for robust domain adaptation under noisy environments. IEEE Transactions on Multimedia. pp. 6157-6170. 2023.

2. Missleading compared methods, the authors compared "noise-resistant baseline DRANet-SWD (Soletal.,2025)" which is only a UDA method, not for noisy UDA. Such comparisons mis-lead readers to think that their method beat SOTA, which is not true. I think such comparision is not appropriate.


3. Line 350: CMAL+EGCA without entropy normalization already achieves 71.5%, which is not consistent with data in Table 2.

**Questions:**

Please refer to the weakness.

---

### Official Review · Reviewer_aZzg · 2025-11-03

**Soundness:** 2
**Presentation:** 1
**Contribution:** 2
**Rating:** 2
**Confidence:** 3

**Summary:**

In this paper, the authors propose an instance- and category-level Reliability Scheduling Framework (RSF) for the unsupervised domain adaptation (UDA) problem under label noise in the source domain. In particular, they introduce a confidence-modulated adaptive learning loss to regulate the participation of uncertain data in source model training, and an entropy-guided confusion alignment via class-wise entropy regularization. The proposed method is validated on several UDA benchmarks, including Office-Home, Office-31 and VisDA.

**Strengths:**

- The manuscript is easy to follow.

- The problem studied is interesting and represents a promising research direction.

**Weaknesses:**

- The presentation of the paper, especially the Introduction, could be further polished.

- There is insufficient evaluation on larger datasets or more challenging benchmarks (e.g., DomainBed [1]).

- Some relevant related work and discussion are missing (see Questions).

- The proposed method appears to be somewhat incremental.


[1] Moment matching for multi-source domain adaptation.

**Questions:**

1. Writing issues in the Introduction

    - In lines 71–74, the authors mention that the CMAL (sample-level) module employs entropy-based confidence estimation, while the EGCA (class-level) module emphasizes low-uncertainty target samples. As both modules depend on uncertainty information, the distinction between them is unclear. This overlap makes the presentation somewhat confusing and weakens the logical flow of the paper.
     - The content in lines 82–92 is overly repetitive and could be substantially condensed.


2. Methodological concerns

    - The basic idea of the CMAL component is to use a specific loss format to reduce the participation of uncertain data in the neural-network update process. However, how do the authors ensure that the uncertain ones are actually the noisy ones? What is the difference between the proposed loss term and other noise-robust losses? See [1] and [2].
    - The difference or similarity between the proposed EGCA component and Do We Really Need to Access the Source Data? Source Hypothesis Transfer for Unsupervised Domain Adaptation (SHOT) [3] should be explicitly discussed.


[1] Early-Learning Regularization Prevents Memorization of Noisy Labels.

[2] Symmetric Cross Entropy for Robust Learning with Noisy Labels.

[3] Do We Really Need to Access the Source Data? Source Hypothesis Transfer for Unsupervised Domain Adaptation.

---

### Note · Authors · 2025-11-14

I have read and agree with the venue's withdrawal policy on behalf of myself and my co-authors.